# Fabrication and Characterization of Core-Shell Electrospun Fibrous Mats Containing Medicinal Herbs for Wound Healing and Skin Tissue Engineering

**DOI:** 10.3390/md17010027

**Published:** 2019-01-05

**Authors:** Elahe Zahedi, Akbar Esmaeili, Niloofar Eslahi, Mohammad Ali Shokrgozar, Abdolreza Simchi

**Affiliations:** 1Department of Chemical Engineering, North Tehran Branch, Islamic Azad University, P.O. Box 19585/936, Tehran, Iran; zahedi.ez@gmail.com (E.Z.); akbaresmaeili@yahoo.com (A.E.); 2Department of Textile Engineering, Science and Research Branch, Islamic Azad University, P.O. Box 14515/775, Tehran, Iran; eslahi_n@yahoo.com; 3National Cell Bank Department, Pasteur Institute of Iran, Tehran 13164, Iran; mashokrgozar@yahoo.com; 4Department of Materials Science and Engineering, Sharif University of Technology, Azadi Avenue, P.O. Box 11365/8639, Tehran, Iran; 5Institute for Nanoscience and Nanotechnology, Sharif University of Technology, Azadi Avenue, P.O. Box 11365/8639, Tehran, Iran

**Keywords:** co-axial electrospinning, core-shell fiber, polysaccharide, Aloe vera extract, wound healing

## Abstract

Nanofibrous structures mimicking the native extracellular matrix have attracted considerable attention for biomedical applications. The present study aims to design and produce drug-eluting core-shell fibrous scaffolds for wound healing and skin tissue engineering. Aloe vera extracts were encapsulated inside polymer fibers containing chitosan, polycaprolactone, and keratin using the co-axial electrospinning technique. Electron microscopic studies show that continuous and uniform fibers with an average diameter of 209 ± 47 nm were successfully fabricated. The fibers have a core-shell structure with a shell thickness of about 90 nm, as confirmed by transmission electron microscopy. By employing Fourier-transform infrared spectroscopy, the characteristic peaks of Aloe vera were detected, which indicate successful incorporation of this natural herb into the polymeric fibers. Tensile testing and hydrophilicity measurements indicated an ultimate strength of 5.3 MPa (elongation of 0.63%) and water contact angle of 89°. In-vitro biological assay revealed increased cellular growth and adhesion with the presence of Aloe vera without any cytotoxic effects. The prepared core-shell fibrous mats containing medical herbs have a great potential for wound healing applications.

## 1. Introduction

Skin is the outer covering of the body, which prevents the infiltration of disease and infections and protects the underlying organs [1]. Physicians use wound healing substitutes or artificial skins to restore wounds caused by burns, trauma, disease, and aging [2,3]. The creation of such a replacement requires a three-dimensional, porous, biocompatible, and preferably biodegradable scaffold, which emulates the skin’s extracellular matrix (ECM) and provides a suitable substrate for cellular adhesion and growth [4].

The selection of the scaffold components plays an influential role in ensuring the success of skin tissue engineering. Different natural and synthetic materials have been used for the fabrication of biomedical scaffolds. The main obstacle in using natural polymers is their poor mechanical properties. On the other hand, although synthetic polymers have superior mechanical properties over natural polymers and can be easily processed, they show lower biological activity [4]. Among different synthetic polymers, polycaprolactone (PCL) has widely been used in tissue engineering applications because of its suitable mechanical properties and biocompatibility [5]. Nevertheless, hydrophilicity of PCL is a major drawback for tissue engineering applications [6,7]. It is well known that hydrophilicity of nanofibrous webs has a significant effect on the cell adhesion and growth, and thus is important in repairing damaged tissues. Therefore, attempts have been made to add biological agents and natural polymers to PCL in order to provide an appropriate substrate for skin regeneration [8]. As the main part of skin consists of keratin as a fibrous protein, this natural biopolymer has been used in the present study. Keratin is a structural element in many living organisms. The presence of intermolecular bonding of disulfide cysteine and inter- and intra- molecular bonding of various amino acids makes keratin a highly stable protein with unique physicochemical properties, which can be used in a wide range of biomedical applications [9]. In recent years, studies have demonstrated the feasibility of keratin electrospinning. Aluigi et al. [10] manufactured keratin/poly(ethylene oxide) (PEO) scaffold by electrospinning. Yuan et al. [11] found that keratin improved cell-material interaction in polylactic acid (PLA)/keratin electrospun fibers for tissue engineering applications.

Chitosan is another natural biopolymer that has been the focus of much research in the past few years. This natural polysaccharide, which is found in crustacean shells such as crabs and shrimp, has been considered extensively for biomedical applications because of its non-toxicity, biocompatibility, biodegradability, high absorption properties, and cost-effectiveness [12]. Over the last few years, chitosan has emerged as a favorable candidate for tissue engineering applications, particularly for wound healing and skin regeneration [13]. Besides biocompatibility and cytocompatibility, chitosan inhibits growth of many types of bacteria, fungi, and yeasts [14]. The bactericidal property of chitosan is of special significance for tissue engineering because utilizing antibacterial agents that may lead to toxicity delays the healing process, or induces pathogen resistance [15]. 

In the present work, nanofibrous mats based on PCL/keratin/chitosan were prepared. It is well known that nanofibers are the most suitable materials for wound healing, owing to their unique features such as high surface-to-volume ratio, high porosity, suitable mechanical properties, and permeability [16]. Electrospinning is a simple, versatile, and efficient technique to prepare fibrous mats from different types of polymers [17]. Recently, processing of hollow and core-shell nanofibers have gained significant attention. These structures with a large surface area to volume ratio can be incorporated with biologically active molecules in order to gain a controlled release profile [18]. In the case of core-shell structure, the shell can be functionalized to impart specific properties while the core keeps the intrinsic properties of the fiber, or vice versa [19]. Therefore, the produced fibers simultaneously benefit from both intrinsic properties of the two polymers. Among different techniques of multi-step template synthesis, surface initiated atom transfer radical polymerization and co-axial electrospinning (or two-fluid electrospinning); the latter is still the most versatile method for fabricating core-shell fibers [20]. When two dissimilar liquids are simultaneously delivered through a co-axial capillary, nanofibers are formed in a high electric field by solvent evaporation and stretching. Studies have shown that processing parameters (feeding ratio, applied electric field, needle diameter, etc.), solution properties (viscosity, surface tension, conductivity, and elasticity), the environment (temperature, humidity, and static electricity), and immiscibility of the liquids affect the size, morphology, and uniformity of the fabricated fibers [21]. This technique has gained much attention in recent years and holds great promise for the preparation of core-shell non-woven scaffolds.

We employed the co-axial electrospinning technique in order to encapsulate Aloe vera extract (core) into the PCL/chitosan/keratin nanofibers (shell). Aloe vera is the most commonly used plant for skin and hair. Aloe vera gel consists of 99% water, while the rest is composed of vitamins (including b-carotene, vitamin E, vitamin C, and B12), antioxidant, enzymes, minerals, sugars, anthraquinones, sterols, salicylic acid, and amino acids [22,23]. This therapeutic herb is very effective in the treatment of many wounds, injuries, burns, insect bites, and acne because of its antimicrobial properties [22]. Aloe vera has wound healing, anti-inflammatory, antibacterial, antiviral, and antioxidant properties [24,25]. Incorporation of Aloe vera in electrospun nanofibers through mixing with the polymer has already been examined [26]. Nevertheless, no control over the release rate can be attained. Therefore, this study attempts to fabricate core (Aloe vera extract)-shell (PCL/keratin/chitosan) nanofibers. The hydrophilicity, mechanical properties, and cytocompatibility of the mats were investigated.

## 2. Materials and Methods

### 2.1. Materials

Polycaprolactone (Mn = 10,000 GPC), chitosan with medium molecular weight (Mw = 190–310 kDa), and polyethylene oxide (M_W_ = 900 kDa) were purchased from Sigma-Aldrich (St. Louis, MO, USA). Aloe vera extract was obtained from Acidgone (Austin, TX, USA). Acetic acid and formic acid (analytical grades) were supplied by Merck (Darmstadt, Germany). Keratin was extracted from wool fibers based on our previous study [27]. Briefly, cleaned defatted wool fibers were mixed with an aqueous solution containing 8 M urea, 0.5 M sodium pyrosulfite, and 0.05 M sodium dodecyl sulfate (SDS) at 65 °C, and stirred for 24 h. After filtration and dialysis, the extracted solution was lyophilized for 48 h by a freeze-dryer (Lyotrap/Plus, Wickford, UK) at −40 °C to obtain Keratin powder (M_W_: 45–65 kDa).

### 2.2. Electrospinning of Fibrous Mats

PCL (8% wt) and chitosan (3% wt) solutions were prepared separately by dissolving the polymers in formic acid and acetic acid solutions, respectively. The solutions were then mixed at a ratio of 30:70 under magnetic stirring for 24 h at room temperature. To decrease the solution viscosity, an aqueous PEO solution (5% wt) was added to the mixture. Finally, different amounts of keratin (0%, 5%, 10%, and 15%) were mixed with the PCL/chitosan/PEO solution. The suitable amount of keratin to attain the finest fiber diameters was determined.

A single jet electrospinning apparatus (Model HVPS-35/500, ANSTCO, Tehran, Iran) was employed to prepare fibrous mats. In order to attain long and uniform fibers, the processing parameters were adjusted by trial-and-error. It was found that a feeding rate of 0.1 mL/min, an applied voltage of 15 kV, and a tip-to-collector distance of 100 mm were suitable.

### 2.3. Fabrication of Core-Shell Fibers

For co-axial electrospinning, Aloe vera extract (20% wt) was mixed with PEO solution (5% wt) and delivered through a co-axial capillary concurrently with the PCL/chitosan/keratin liquid. We have experimentally found that a feed rate of 0.1 mL/min, an applied voltage of 16 kV, and a tip-to-collector distance of 120 mm could yield a uniform core-shell structure.

### 2.4. Characterization of Fibrous Mats

The fabricated fibrous mats were gold coated by sputtering and examined using a field emission scanning electron microscope (FESEM, Philips Model XL30, Eindhoven, The Netherland) at an accelerating voltage of 30 kV. The mean diameter of the fibers was analyzed by measuring the diameter of 50 random fibers from FESEM images. An image analysis software (Image J, National Institute of Health, Bethesda, MD, USA) was utilized to determine the fiber size distribution. A JEOL transmission electron microscope (TEM, Zeiss, EM10C, Berlin, Germany) was used at 100 kV voltage to analyze the core-shell structure.

Fourier transform infrared (FT-IR) spectroscopy (AVATAR, Thermo, Waltham, MA, USA) of the mats was performed in transmission mode using KBr pellets in the wavenumber range of 4000–400 cm^−1^. Mechanical properties of the scaffolds were assessed by a universal tensile testing machine (AG-5000G, Shimadzu, Kyoto, Japan) according to ASTM D638 at a strain rate of 5 mm/min and 2 cm gauge length. Water contact-angle goniometry (OCA 15 Plus, Dataphysics, Filderstadt, Germany) was employed to evaluate the wettability of the fibrous membranes. Triplicate samples (*n* ≥ 3) were examined and the average values were reported. 

### 2.5. In Vitro Assays

#### 2.5.1. Cell Culture

L929 (NCBI C161) cells were obtained from the cell bank of Pasteur Institute of Iran. After defreezing, the cells were transferred to a flask containing Roswell Park Memorial Institute (RPMI) culture medium with 10% fetal bovine serum (FBS). The flask was then located in an incubator at 37 °C with an atmosphere of 90% moisture and 5% oxygen.

#### 2.5.2. MTT Assay

The dimethyl-thiazole diphenyl tetrazolium bromide (MTT, Sigma, USA) assay was employed as an indirect method to determine the cell viability. The specimens were first exposed to an ultraviolet irradiation for 45 min for sterilization. The extraction process was carried out based on ISO 10993-5, in which 1 mL of the culture medium was added to the specimens with 6 cm^2^ surface area. A specified amount of the culture medium (RPMI) was considered as control.

For quantitative toxicity assessment, 1 × 10^4^ cells along with 100 μL culture medium were added into each well of a 96-well culture plate and then incubated at 37 °C overnight. After ensuring cell adhesion, the culture medium on the cells was removed and 90 μL of specimens extract along with 10 μL of FBS were added to each well for another 24 h. MTT dissolved in PBS and then 100 μL of this solution at a concentration of 0.5 mg/mL was added to each well and incubated for 4 h. After removal of the MTT solution, isopropanol was added to dissolve the resulting purple crystals for 15 min. Finally, the absorbance was calculated using an Elisa Reader (STAT FAX 2100, St. Louis, MO, USA) at a wavelength of 545 nm. The cell viability was measured in comparison with the control.

#### 2.5.3. Cell Adhesion

The sterilized specimens were placed in each well of a six-well tissue culture plate. Then, the cells with a density of 25 × 10^3^ cells/cm^2^ were seeded on the samples and incubated at 37 °C in a humidified atmosphere with 5% CO_2_ for 4 h. After cell adhesion, a certain amount of culture medium containing 10% FBS was added to each well. The culture medium on specimens was removed after one day and washed with phosphate buffered saline (PBS) for 30 s. The cells were fixed with 3.5% glutaraldehyde and then dehydrated with increasing concentrations of ethanol. Finally, cell adhesion to the specimens was investigated by scanning electron microscopy (FESEM, MIRA3TESCAN-XM, Brno, Czech Republic)

## 3. Results and Discussion

### 3.1. Morphological and Dimensional Characterizations of Electrospun Fibers

At first, we examined the electrospinning ability of the chitosan/PCL blend. As Figure 1a shows, tiny droplets were formed, which indicated instability of the process. After adding 5% PEO, long and uniform fibers with an average diameter of 172 ± 48 nm were formed (Figure 1b). PEO decreases the viscosity of the solution [28], and thus improves the spinnability of the backbone polymer. Afterwards, the effect of keratin addition on the morphology and size of the fibers was studied. Figure 1c–e show representative SEM images of chitosan/PCL fibers containing different amounts of keratin (5%, 10%, and 15%). It was found that keratin addition increased the average diameter of the fibers. Additionally, more beds were formed. This finding could be attributed to the effect of keratin on the viscosity [29,30,31,32]. To have the finest possible fibrous structure, fibers containing 5% keratin were selected and used for further experiments.

### 3.2. Co-Axial Electrospinning of Core-Shell Fibers

Before fabrication of core-shell fibers, electrospinning of the PEO/Aloe vera mixture was performed to ensure the ability of the core to be electrospun. Figure 2a shows that long and continuous fibers are formed by electrospinning of PEO solution containing the medicinal herb. Figure 2b,c illustrate SEM images of core-shell nanofibers (PCL/chitosan/keratin as the shell and PEO/Aloe vera as the core) and their physical mixture, respectively. The fiber diameter distribution plots indicated that the average diameter of fibers for core, core-shell, and physical mixture are 196 ± 49, 209 ± 47, and 261 ± 94, respectively. It thus appears that the average diameter of the core-shell fibers is bigger than both the core and the physical mixture. This finding supports the formation of core-shell structure, as shown by others [33]. Meanwhile, a slight degree of fiber adhesion and formation of a spider-web-like morphology were observed when a physical mixture was utilized (Figure 2c). It seemed that physical mixing of the polymeric solutions/suspensions affected the morphology because of a change in the evaporation rate of the solvent [34].

The core-shell structure of the scaffold was further examined by TEM. As Figure 3 shows, the diameters and thicknesses of the core and the shell are 62 nm and 91 nm, respectively. In co-axial electrospinning setup, two dissimilar solutions are delivered by a syringe pump and flow through capillaries. The needle charges with a high voltage and when the electric field is potent enough, a jet of polymer is drawn from the needle and attracted to a grounded collector [21]. As a result, a core-shell structure is attained.

### 3.3. FTIR Analysis

FT-IR spectrum of the prepared mats is shown in Figure 4. The spectrum of PCL/chitosan/PEO (Figure 4a) exhibits a broad absorption band in the range of 3400–3500 cm^−1^, which is assigned to N–H and O–H stretching vibrations of chitosan. The peaks at about 1000–1300 cm^−1^ are related to C–O stretching vibration of ether groups in PEO and PCL, as well as in chitosan [35]. The intense bands around 2865 and 2990 cm^−1^ are related to the symmetric and asymmetric stretching vibrations of CH_2_ aliphatic. The stretching vibration of carbonyl group (C=O) of PCL is also observed at 1720 cm^−1^ [36]. The peaks at about 1560 cm^−1^ and 1060 cm^−1^ are ascribed to N–H bending of amide II and C–O–C stretching vibration of glycoside linkage in chitosan, respectively [37]. In the presence of keratin, two peaks at around 1630 cm^−1^ (amide I carbonyl stretching vibration) and 1280 cm^−1^ (C–N stretching vibration) appeared (Figure 4b). The amide II also falls at 1540 cm^−1^, which is related to N–H bending and C–H stretching vibration. The spectrum of Aloe vera/PEO (Figure 4c) exhibits an absorption peak at 3324 cm^−1^, which corresponds to O–H vibration of phenolic groups in the structure of Aloe vera. Stretching vibrations of aromatic ring (C=C) are observed in the range of 1400–1600 cm^−1^. The characteristic band of carboxyl groups at around 1596 cm^−1^ is referred to as COO^−^ stretching [38]. The spectrum illustrated in Figure 4d confirms the presence of all components in the core-shell nanofibers. The intensity and location of some peaks such as COO^−^ are altered by adding Aloe vera because of possible intermolecular interactions between polymers.

### 3.4. Physicomechanical Properties of Electrospun Nanofibers

The skin on each part of the body has unique mechanical properties matching the function of the respective part [39,40,41,42]. Figure 5 shows typical stress–strain curves of electrospun mats under tensile loading. Measured values of ultimate tensile strength and elongation are reported in Table 1. The energy required to break the samples, as an indicator of toughness, is also presented. In agreement with a previous study [10], the addition of keratin slightly improves the strength at the expense of ductility. This finding can be attributed to a reduction in the motion of polymer chains. Possible intermolecular hydrogen bonding and polar interactions between the components (e.g., between hydroxyl groups of Aloe vera, ester functional groups of PCL, carbonyl groups of keratin, and chitosan amino groups) also affected the strength. Interestingly, the core-shell fibrous structure exhibits better mechanical properties in comparison with their physical mixture. Toughness and elongation were significantly improved (from about 0.5 J/m^3^ to 2.3 J/m^3^ and 30% to 63%, respectively) as a result of interactions between hydroxyl, amino, and carbonyl groups of the polymers.

The water contact angle test was employed to evaluate the hydrophilicity of electrospun mats. The results are presented in Figure 6. Keratin reduced the hydrophilicity of the PCL/chitosan mat as this natural fibrous protein has both hydrophilic (Threonine and Serine) and hydrophobic (Phenylalanine, Alanine, Cysteine, Valine, Leucine, Isoleucine, and Tyrosine) chains and its poor wettability is due to the cross-linking of disulfide cystine in the structure [43]. On the other hand, PEO is strongly hydrophilic and almost soluble in water. Therefore, the PEO/Aloe vera mat exhibits a low water contact angle. The core-shell structure exhibits similar hydrophilicity to the shell polymer, while the physical mixture has the highest water contact angle.

### 3.5. In Vitro Cell Study

Biocompatibility of a material is not an intrinsic property, it is a result of complex interactions between a scaffold and its surrounding tissue [44]. The hydrophilicity of scaffolds along with the appropriate surface topography are essential requirements for the biocompatibility. Additionally, the existence of specific functional groups in fibers has an influential role in cell adhesion to the surface [4,45]. Representative SEM images of fibroblast cells grown on the nanofibers are shown in Figure 7. Although SEM is not a well-suited method for quantification of cell adhesion, it seems that more cells are adhered to the PCL/chitosan mat (Figure 7a) than to the PCL/chitosan/keratin (Figure 7b). This difference is most probably because of the higher hydrophilicity of the former mat (Figure 6). The presence of extracellular secretions and multiple filopodia extended from the cells to the substrate could be a good indication of cells interactions with the surrounding fibers because of the large surface area available for cellular attachment [46,47].

Figure 7c indicates that the PEO/Aloe vera mat provides adhesive sites for cell attachment and facilitates spreading of the cells on its surface. Previous studies have shown that Aloe vera encourages cell migration, proliferation, and growth [48]. The core-shell structure shows more adhesive cells along with extracellular secretions (Figure 7d). Some studies have shown that surface topography as well as fiber structure affect cell attachment and spreading [49,50].

MTT assay was performed to evaluate cytotoxicity of nanofibrous scaffolds. The results are presented in Figure 8. The cell viability is >80% after 72 h incubation, indicating that the mats are not cytotoxic. The addition of keratin to PCL/chitosan led to a slight decrease in the cytocompatibility. However, there is no significant difference between PCL/chitosan/keratin and the core-shell scaffold. PCL, chitosan, and keratin are biocompatible and biodegradable polymers that have been widely studied [31,51]. Aloe vera contains significant growth factors, which aid cell proliferation and differentiation [52]. As the fabri cated scaffolds are nontoxic and provide adequate support for cell growth, they have the potential to be used as substitute materials for wound dressing and skin tissue engineering.

## 4. Conclusions

Core-shell electrospun fibers containing medicinal herbs (Aloe vera extract) were fabricated by the co-axial electrospinning technique. The core is composed of PEO/herb with a diameter of ~209 nm. The shell consisted of PCL/chitosan/keratin with a thickness of ~91 nm. As the PEO/herb core is unstable in the physiological media, the shell protected the core, allowing to incorporate the herb in the membrane. Mechanical testing indicated that the core-shell fibrous mats were stronger than the polymer shell. The tensile strength was about 5 MPa. Interestingly, the elongation was significantly improved by the co-axial electrospinning. It was also found that keratin slightly increased the tensile strength, but reduced the elongation. In vitro cell studies supported biocompatibility of the mats. Improved cell adhesion on the core-shell structure was also noticed. Therefore, the core-shell fibrous mats have a great potential to be used for wound healing and skin tissue engineering.

## Figures and Tables

**Figure 1 marinedrugs-17-00027-f001:**
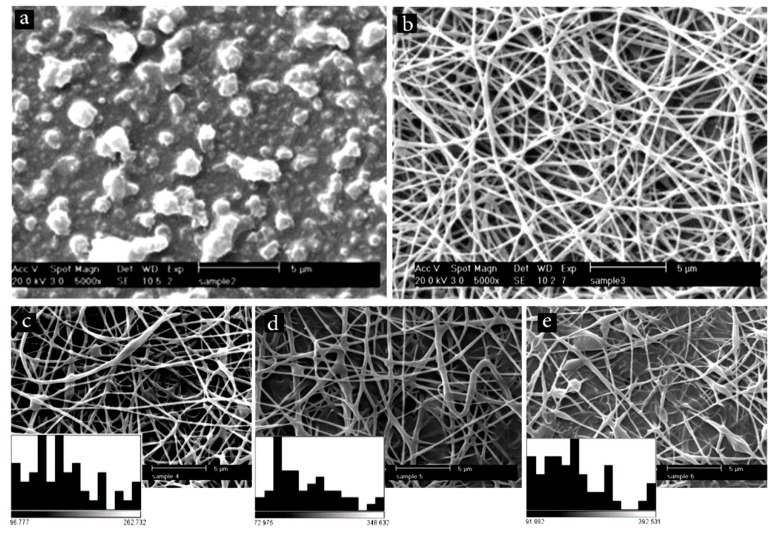
Scanning electron microscope (SEM) images showing the morphology of electrospun polymers: (**a**) chitosan/polycaprolactone (PCL); (**b**) chitosan/PCL containing 5% poly(ethylene oxide) (PEO), shows the effect of PEO addition; (**c**–**e**) show the effect of keratin concentration (5%, 10%, and 15%, respectively) on the morphology and size distribution of chitosan/PCL nanofibers. The inset histograms in (**c**,**d**) show size distribution of the fibers.

**Figure 2 marinedrugs-17-00027-f002:**
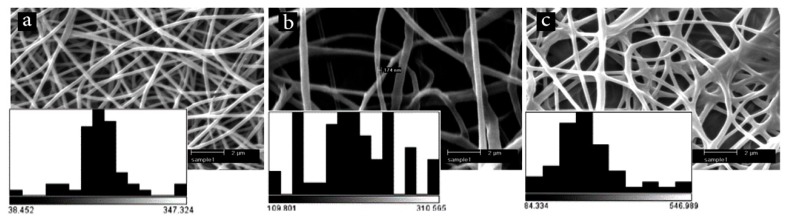
SEM images and corresponding fiber diameter distribution plots for electrospun nanofibers: (**a**) core, (**b**) core-shell, and (**c**) physical mixture of core and shell.

**Figure 3 marinedrugs-17-00027-f003:**
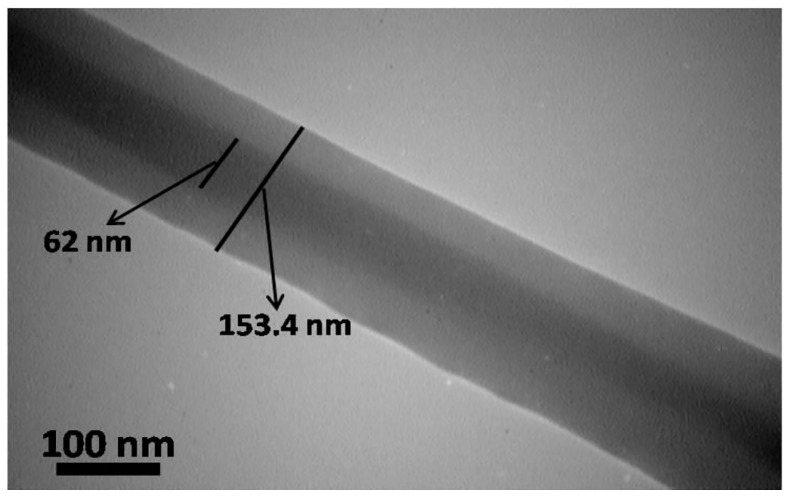
Transmission electron microscope (TEM) image of a core-shell nanofiber prepared by co-axial electrospinning.

**Figure 4 marinedrugs-17-00027-f004:**
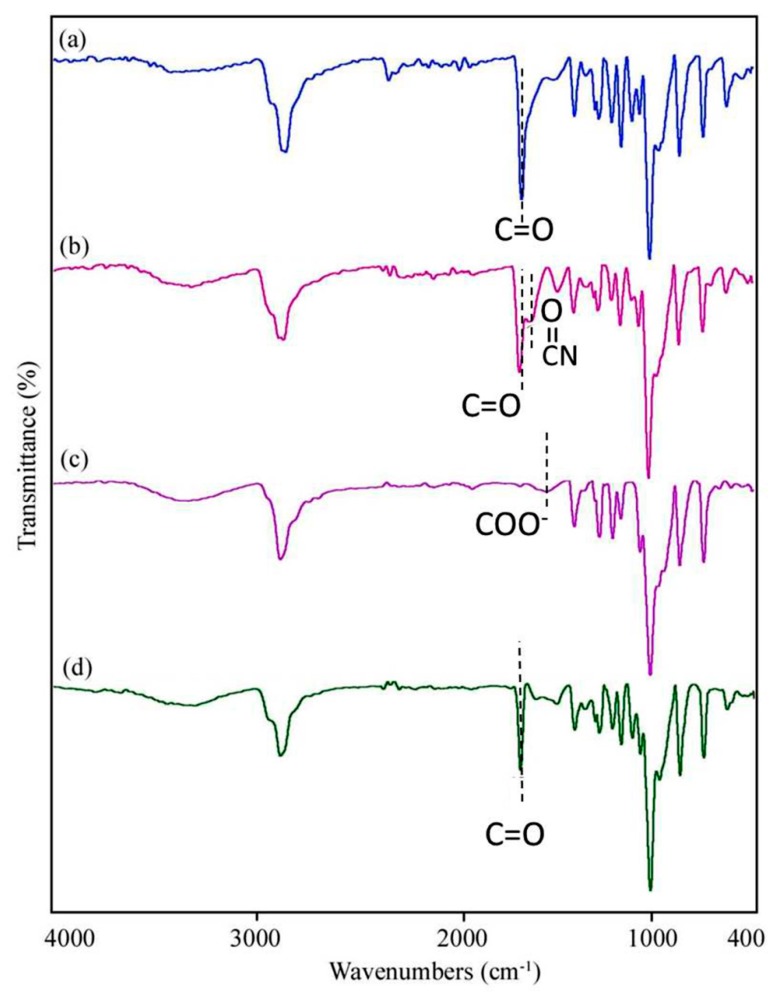
Fourier transform infrared (FT-IR) spectrum of (**a**) PCL/chitosan/PEO, (**b**) PCL/chitosan/PEO-keratin, (**c**) Aloe vera/PEO, and (**d**) core-shell fibers.

**Figure 5 marinedrugs-17-00027-f005:**
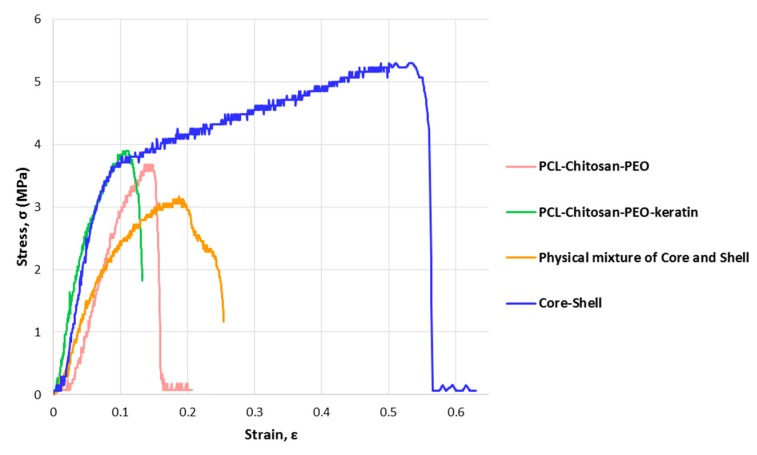
Typical tensile stress-strain curves of fibrous mat showing enhanced mechanical properties of the core-shell structure containing the medicinal herb.

**Figure 6 marinedrugs-17-00027-f006:**
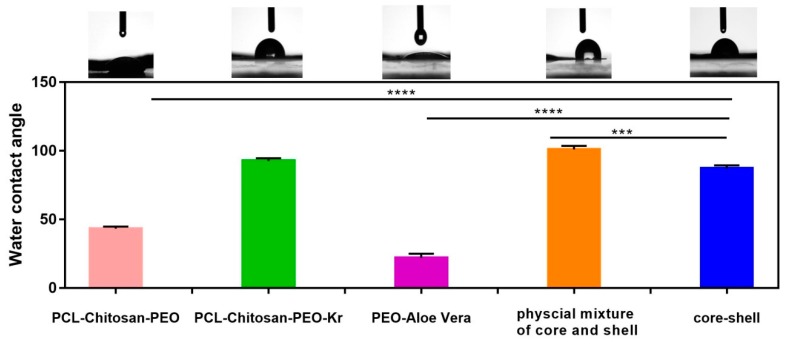
Effect of composition, morphology, and method of processing on the hydrophilicity of fibrous mats as indicated by water contact angle measurement. Error bars indicate standard error of the means, asterisks mark significance levels of *p* < 0.001 (***) and *p* < 0.0001 (****) (*n =* 3).

**Figure 7 marinedrugs-17-00027-f007:**
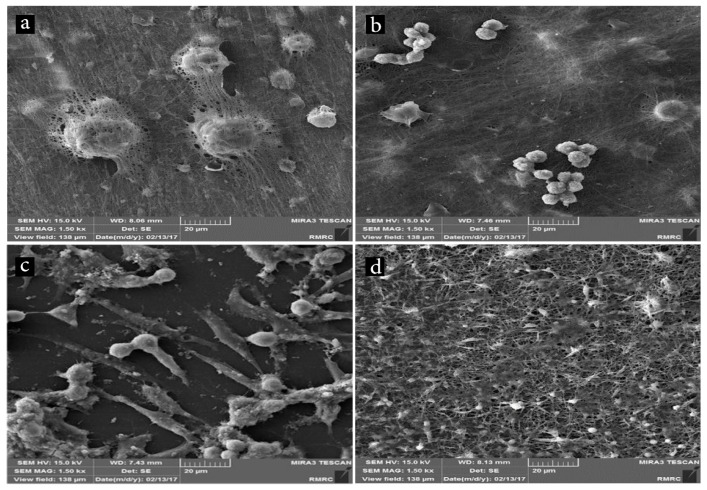
SEM images showing cell adhesion on the surface of (**a**) PCL/chitosan, (**b**) PCL/chitosan/keratin, (**c**) Aloe vera/PEO core, and (**d**) core-shell fibrous mats.

**Figure 8 marinedrugs-17-00027-f008:**
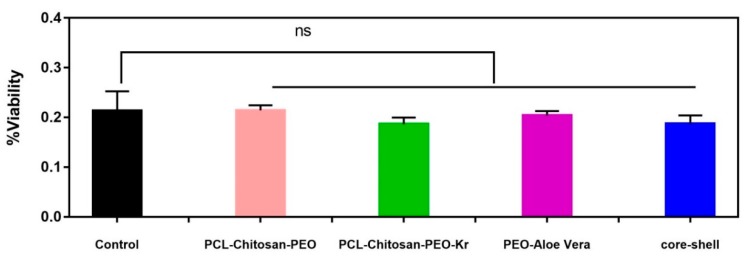
Cell viability of different fibrous mats as assessed by dimethyl-thiazole diphenyl tetrazolium bromide (MTT) after 72 h incubation. Error bars indicate standard error of the means (*n =* 3). *ns* is defined as a non-significant difference.

**Table 1 marinedrugs-17-00027-t001:** Mechanical properties of electrospun mats as measured by tensile testing (*n =* 3). PCL—polycaprolactone; PEO—poly(ethylene oxide).

Sample	Tensile Strength (MPa)	Elongation at Break (%)	Toughness (J/m^3^)
PCL/Chitosan/PEO	3.6 ± 0.1	21 ± 2	0.31 ± 0.2
PCL/Chitosan/PEO/Keratin	3.9 ± 0.3	10 ± 4	0.34 ± 0.2
Physical mixture of polymers	3.2 ± 0.5	30 ± 5	0.54 ± 0.3
Core-shell	5.3 ± 0.4	63 ± 8	2.34 ± 0.2

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
