# Peer review of "Fabrication and Characterization of Core-Shell Electrospun Fibrous Mats Containing Medicinal Herbs for Wound Healing and Skin Tissue Engineering"

_marinedrugs, 2019, doi:10.3390/md17010027_

Round 1

Reviewer 1 Report

Comments and suggestions for Authors:

This manuscript aims at the design and production of 
drug-eluting core-shell fibrous scaffolds for wound healing and skin tissue engineering. 
However, the results presented do not evidence sufficient information for this hypothesis.

This manuscript in its actual state it is not publishable in this journal or other relevant journals in the field. The English need extensive revision. I advise that the authors should request help for this.

In the introduction concurrent electrospinning 
is not described, and it is mentioned in the rest of the manuscript. The authors should explain in what relies on this technique.

Materials and methods:

.                 “The extraction process was carried out based on the ISO 10993-5, in 
137, which 1 ml of the culture medium was added to each sterile specimen per 66 cm2 area”
. The ISO correspondent to the procedure to be used to perform extract is the ISO 10993-12, and the area is defined based on the thickness of the materials to be tested. For example for a material with 0,5 mm thick, the area is 6 cm2, not 66 cm2. Also, sentences such as “environment was removed and added to the cells after 3 days” are wrong. What is the environment? Culture media??

Regarding cell adhesion studies, what was the rationale to use 20000-30000 cells per well? It is not referred to the size of the sample, so it is not clear why this cell density was used.

.                 “The culture medium on specimens was removed after 1 day and washed with phosphate buffered saline (PBS) for 30 s. Then, 3.5% glutaraldehyde was used for cell fixation in a refrigerator 
for 2 hours and washed with 40, 60, 80, and 100% alcohol.” This sentence is not clear and not accurate. The authors probably wanted to write that cells were fixed with 3.5% glutaraldehyde and then dehydrated with increasing concentrations of ethanol (not washed).

.                 Results and discussion:

.                 In figure 1 caption it is not described what are the figures above SEM pictures.

.                 Line 198, letter size is increased in some words.

Figure 5: Y axis, change Mpa to MPa; caption of this figure has comments. This is not correct.

Table 1: the values presented, such as 3/6 or 0/31 are not clear. What is this? Where are the standard deviations of the results? How many samples were tested per condition?

.                 Figure 6: Where are the standard deviations of the results? How many samples were tested per condition?

.                 In vitro cell study: “It was found that more cells were adhered to the PCL/chitosan mat” or “Fig. 7c indicates that more cells are attached to the surface of the PEO/Aloe Vera mat. “ The authors cannot state this from SEM pictures.

.                 “Some studies have shown that 
surface topography as well as fiber structure effect cell attachment on the electrospun fibers”. The english language is not suitable.

“The results presented in Fig. 8 determine acceptable cell viability (>80%)”. The authors should revise this.

“However, there is no significant difference between PCL/chitosan/keratin and the core-shell scaffolds.” The authors cannot say this, because they did not perform any statistical evaluation.

Conclusions:

.                 “Since PEO/herb core is 
unstable in the physiological media, the shell protected the core letting to incorporate the herb in the 
membrane with more controlled release.” 
First, where is the release study? The title of the paper is not supported by the results.

Author Response

1) This manuscript aims at the design and production of 
drug-eluting core-shell fibrous scaffolds for wound healing and skin tissue engineering. However, the results presented do not evidence sufficient information for this hypothesis.

Response: Thank you for taking time to review the manuscript. In light of your and other’s suggestions and comments, the manuscript was revised. We hope that the state of the paper would be acceptable for publication.

2) This manuscript in its actual state it is not publishable in this journal or other relevant journals in the field. The English need extensive revision. I advise that the authors should request help for this.

Response: We are sorry for language weaknesses. None of the authors is native English. We tried our bests to improve the language.  

3) In the introduction concurrent electrospinning is not described, and it is mentioned in the rest of the manuscript. The authors should explain in what relies on this technique.

Response: We agree with you. The technique was explained in “Introduction” (see PP. 4-5). Also, we highlighted the advantages of this technique over common electrospinning.

4) “The extraction process was carried out based on the ISO 10993-5, in 
137, which 1 ml of the culture medium was added to each sterile specimen per 66 cm2 area”
. The ISO correspondent to the procedure to be used to perform extract is the ISO 10993-12, and the area is defined based on the thickness of the materials to be tested. For example for a material with 0,5 mm thick, the area is 6 cm2, not 66 cm2. Also, sentences such as “environment was removed and added to the cells after 3 days” are wrong. What is the environment? Culture media??

Response: The surface area was corrected. This section in “Materials and methods” was revised for clarification.

5) Regarding cell adhesion studies, what was the rationale to use 20000-30000 cells per well? It is not referred to the size of the sample, so it is not clear why this cell density was used.

Response: The size of the samples was stated.

6) “The culture medium on specimens was removed after 1 day and washed with phosphate buffered saline (PBS) for 30 s. Then, 3.5% glutaraldehyde was used for cell fixation in a refrigerator 
for 2 hours and washed with 40, 60, 80, and 100% alcohol.” This sentence is not clear and not accurate. The authors probably wanted to write that cells were fixed with 3.5% glutaraldehyde and then dehydrated with increasing concentrations of

Response: The sentence was rewritten for clarification.

7) In figure 1 caption it is not described what are the figures above SEM pictures.

Response: The figure caption was improved.

8) Line 198, letter size is increased in some words.

Response: Corrected.

9) Figure 5: Y axis, change Mpa to MPa; caption of this figure has comments. This is not correct.

Response: Corrected and revised.

10) Table 1: the values presented, such as 3/6 or 0/31 are not clear. What is this? Where are the standard deviations of the results? How many samples were tested per condition?

Response: Corrections were made. Standard deviation and number of tests were indicated.

11) Figure 6: Where are the standard deviations of the results? How many samples were tested per condition?

 Response: The standard deviation was indicated in the graph. n=3 is mentioned in the caption.

12) In vitro cell study: “It was found that more cells were adhered to the PCL/chitosan mat” or “Fig. 7c indicates that more cells are attached to the surface of the PEO/Aloe Vera mat. “ The authors cannot state this from SEM pictures.

 Response: The sentence was revised for clarification.

13) “Some studies have shown that 
surface topography as well as fiber structure effect cell attachment on the electrospun fibers”. The english language is not suitable.

Response: The sentence was revised.

14) “The results presented in Fig. 8 determine acceptable cell viability (>80%)”. The authors should revise this.

Response: Revised.

15) “However, there is no significant difference between PCL/chitosan/keratin and the core-shell scaffolds.” The authors cannot say this, because they did not perform any statistical evaluation.

 Response: Statistical data was presented in Fig. 8. The sentence was revised.

16) “Since PEO/herb core is unstable in the physiological media, the shell protected the core letting to incorporate the herb in the membrane with more controlled release.” 
First, where is the release study? The title of the paper is not supported by the results.

Response: Thank you for this comment. We agree with you that the drug release should be monitored. Since Aloe Vera does not show distinct peaks in UV-Vis spectroscopy, we could not determine the release profile. HPLC can be used which we do not have access to. There, we revised the sentence and title to avoid confusion.

Reviewer 2 Report

Dear authors, 

Overall, the manuscript is well-written and in my opinion,  the topic is quite interesting for the readers on Pharmaceutical Science. However, there are some issues  to consider before publication. 

- English grammar and spelling should be carefully revised. Some sentences should be rephrased such as line 24 in the abstract (page 1). Line 31, the symbol for 89 degrees in the contact angle should be placed appropriately.

-The quality of some of the figures should be improved and also explained in more detailed. for example, the legend of the SEM in Figures 1 and 2 can barely be read. The bottom histograms in Figure 1 and 2 are not explained anywhere. Please, include a description about the meaning of those figures otherwise remove them. In figure 4, it would be interesting if author label the peaks within the figure and assign the bending or stretching of the molecular motifs so facilitate the comprehension to the readers. 

- In table 1, it is necessary to explain in more detail the meaning of the numbers, what does it mean 3/6 or 0/31?

-Finally, it has not been investigated at all the Aloe Vera release from the mats. Taking into account that  authors are using PCL that it is a polymer for controlled release, it would be interesting to perform some release studies or in vitro permeation assays using Franz cells to understand the release profile of the Aloe Vera from the mats. Some healing experiments would be also great but I understand it may take longer to perform them.

Author Response

1) Overall, the manuscript is well-written and in my opinion, the topic is quite interesting for the readers on Pharmaceutical Science. However, there are some issues to consider before publication. 

Response: Thank you for encouraging. In light of your comments, we improved the manuscript.

2) English grammar and spelling should be carefully revised. Some sentences should be rephrased such as line 24 in the abstract (page 1). Line 31, the symbol for 89 degrees in the contact angle should be placed appropriately.

Response: We are sorry for language weaknesses. We improved writing English.

-3) The quality of some of the figures should be improved and also explained in more detailed. for example, the legend of the SEM in Figures 1 and 2 can barely be read. The bottom histograms in Figure 1 and 2 are not explained anywhere. Please, include a description about the meaning of those figures otherwise remove them. In figure 4, it would be interesting if author label the peaks within the figure and assign the bending or stretching of the molecular motifs so facilitate the comprehension to the readers. 

Response: The quality of figures was improved. The captions were revised for clarification. Peaks in Figure 4 were assigned.

4) In table 1, it is necessary to explain in more detail the meaning of the numbers, what does it mean 3/6 or 0/31?

Response: The values were corrected.

5) Finally, it has not been investigated at all the Aloe Vera release from the mats. Taking into account that authors are using PCL that it is a polymer for controlled release, it would be interesting to perform some release studies or in vitro permeation assays using Franz cells to understand the release profile of the Aloe Vera from the mats. Some healing experiments would be also great but I understand it may take longer to perform them.

Response: Thank you for this valuable comment. We agree with you that determining the release profile and testing healing activity improve the quality of the work. In fact, we could not trace Aloe Vera release by UV-Vis spectroscopy as the herb does not show distinct absorption peaks. HPLC and Franz Cells are not also available and accessible to us.  Healing experiments need in vivo animal assays which are very time consuming and costly. We added some sentences in order to highlight the importance of these tests for future investigations.

Reviewer 3 Report

The manuscript contains  interesting experimental data.

In Introduction the information about current state-of-the-art sometimes was obsolete.Ref.7 is relevant, but review article by Khalf and Madihally.[2017] Recent advances in multiaxial electrospinning for drug delivery. Eur J Pharm Biopharm, 112, 1-17 is more recent and published 6 years later than Ref.7.

Ref.14 and 15 were published in 2007 and 2008. More recent data can be found in latest reviews, e.g. Hamedi et al [2018] Chitosan  based hydrogels and their applications for drug delivery in wound dressings..A review. Carbohydrate Polym 199, 445 - 460 or Chaudhury et al [2016] Future prospects for scaffolding methods and materials in skin tissue engineering.A review.Int. J. Mol. Sci. 201617(12), 1974. The mistakes in Ref 10 must be corrected. The journals titles are not indicated in most of the references.

Author Response

1) The manuscript contains interesting experimental data.

Response: Thank you for your positive opinion.

2) In Introduction the information about current state-of-the-art sometimes was obsolete.Ref.7 is relevant, but review article by Khalf and Madihally.[2017] Recent advances in multiaxial electrospinning for drug delivery. Eur J Pharm Biopharm, 112, 1-17 is more recent and published 6 years later than Ref.7.

Response: The references were updated.

3) Ref.14 and 15 were published in 2007 and 2008. More recent data can be found in latest reviews, e.g. Hamedi et al [2018] Chitosan  based hydrogels and their applications for drug delivery in wound dressings..A review. Carbohydrate Polym 199, 445 - 460 or Chaudhury et al [2016] Future prospects for scaffolding methods and materials in skin tissue engineering.A review.Int. J. Mol. Sci. 2016, 17(12), 1974. The mistakes in Ref 10 must be corrected. The journals titles are not indicated in most of the references.

Response: The references were updated.

Round 2

Reviewer 1 Report

The comments are in the attached file.

Author Response

December 15, 2018

Marine Drug

Dear Editor:

I am writing you concerning the above manuscript which was revised according to the referees’ comments. In reprisal process, one of the reviewer has not been satisfied and insisted on his/her comments. I am writing a rebuttal letter to his/her comments. I hope that you evaluate the manuscript and reconsider the decision.

Should I do more, please let me know.

Sincerely,

A. Simchi

------------------------------------------------------------------------------------------------

1) This manuscript in its actual state it is not publishable in this journal or other relevant journals in the field. The English need extensive revision. I advise that the authors should request help for this. The authors cannot be sorry for language weakness, and being not native is not an excuse to have well written manuscript. I am not as well.

Response: We tried our bests to improve the language of the manuscript by help of others. We honestly mentioned that barely native English speakers can be find in whole country! In contrast of the reviewer’s comment, we believe that the state of the manuscript with regard to language is not so bad to restrict publication in this journal.

2) “The extraction process was carried out based on the ISO 10993-5, in 
137, which 1 ml of the culture medium was added to each sterile specimen per 66 cm2 area”
. The ISO correspondent to the procedure to be used to perform extract is the ISO 10993-12, and the area is defined based on the thickness of the materials to be tested. For example for a material with 0,5 mm thick, the area is 6 cm2, not 66 cm2. Also, sentences such as “environment was removed and added to the cells after 3 days” are wrong. What is the environment? Culture media??

Response: As the referee found out in the first version of the manuscript, there was a typos error about the surface area of the mats (66 cm^2 instead of 6 cm^2). This is simply a typos mistake and NOT a copy and paste of your sentence! Please kindly read “Materials and methods”. The required information was briefly given. No need to give extra information about a standard test which is available online.

3) Regarding cell adhesion studies, what was the rationale to use 20000-30000 cells per well? It is not referred to the size of the sample, so it is not clear why this cell density was used.

Response: In this experiment, the cell density was 25000 cell/cm^2 not the absolute value of cell numbers! We revised the sentence for clarification. Please kindly mind that, the cell density used in this work is common for evaluation of cell attachment, just as an example:

https://www.sciencedirect.com/science/article/pii/S0142961212006539?via%3Dihub

Obviously, MTT was dissolved in PBS (as now mentioned in the revised manuscript).

4) “Since PEO/herb core is unstable in the physiological media, the shell protected the core letting to incorporate the herb in the 
membrane with more controlled release.” 
First, where is the release study? The title of the paper is not supported by the results.

Response: As we agreed with the referee in the first round of reviewing, studying the drug release is interesting. However, did not go through this object because of the following reasons:

·         The main aim of this research focuses on the synthesis of core/shell structure and investigation the mechanical properties and cell spreading (not specifically “drug-eluting” application). We suggest that due to the core shell structure and presence of PCL, better control over release of the herbal material could be achieved.  As the result shows, Aloe Vera increased affinity of cells to attach and spread on the surface of the fibers, while the core and shell individually don’t show this ability. Additionally, based on our mechanical test the core-shell structure represents higher mechanical strength and extensibility which is the most important parameters to wound healing application.

·         As mentioned in “Introduction”, Aloe Vera contains 99% water and 1% other materials such as many vitamins (including b-carotene, vitamin E, vitamin C and B12), antioxidant, enzymes, minerals, sugars, anthraquinones, sterols, salicylic acid, and amino acids, which make it complex material to release study using UV spectrophotometry; for example see:

https://www.ncbi.nlm.nih.gov/pmc/articles/PMC3712247/

https://www.ncbi.nlm.nih.gov/pubmed/26529192

·         Please kindly mind that the time given for revision was only 3 days and performing drug-eluting tests need at least two months! On the other hand, we do not have access to HPLC, thereby we put stress on the synthesis of the core/shell fibers as the main aim of the manuscript and investigation of the properties (rather than drug release studies).